# A New Ignition Source for the Determination of Safety Characteristics of Gases

Stefan H. Spitzer [1,2,*] , Gerald Riesner [1], Sabine Zakel [1] and Carola Schierding [1]

1   National Metrology Institute (PTB), Bundesallee 100, 38116 Braunschweig, Germany
2   Bundesanstalt für Materialforschung und -Prüfung, Unter den Eichen 87, 12205 Berlin, Germany
*   Correspondence: stefan.spitzer@ptb.de; Tel.: +49-531-592-3713

**Abstract:** Safety characteristics are used to keep processes, including flammable gases, vapors, and combustible dusts, safe. In the standards for the determination of safety characteristics of gases and vapors, the induction spark is commonly used. However, classic transformers are hard to obtain, and replacement with new electronic transformers is not explicitly allowed in the standards. This article presents the investigation of five gases that are normally used to calibrate devices for the determination of safety characteristics, the maximum experimental safe gap (MESG), with an electronic transformer, and the values are compared to the ones that are obtained with the standard transformer. Additionally, calorimetric measurements on the net energy of both ignition sources were performed as well as open-circuit voltage measurements. It is concluded that the classic type of transformer can be replaced by the new type obtaining the same results for the MESG and introducing the same amount of energy into the system.

**Keywords:** safety characteristics; maximum experimental safe gap (MESG); ignition source; calorimetry





## 1. Introduction

In the standards for the determination of the safety characteristics of gases, the induction spark is the common ignition source. This is listed in many standards, such as the American and European standard for the determination of explosion limits and limiting oxygen concentration [1–3], the international standards for the determination of the explosibility of gases [4], and the international standards for the determination of the explosibility of dusts [5]. Furthermore, this is also stated in the international standard for the determination of fire potential and oxidizing ability for the selection of cylinder valve outlets [6], the European standard for the determination of the explosion points of flammable liquids [7], and of the maximum explosion pressure and the maximum rate of pressure rise of gases and vapors [8]. Finally, it is also listed in a withdrawn American standard test method for dust explosions [9].

The features of the induction spark, consisting mainly of a transformer and two electrodes, are described briefly in the standards:

- Secondary voltage: 12 kV to 16 kV
- Current: 20 mA to 30 mA
- Electrode distance: 4 mm to 6.35 mm (=1/4 inch)
- Spark duration: 200 ms to 500 ms or continuously (only [5,9]).

No features of the cables are stated. The power is not stated in some of the standards [4,9]; in others, it is stated directly with 200 W [5] or indirectly by stating an energy of 10 J per spark [6]. This leads to a power of 1000 W at 50 Hz or 1200 W at 60 Hz if a spark is generated at every positive and negative halfwave of the sine-shaped voltage. In other standards, the power is stated with 10 W [1,7,8,10] even though the other features seem to be the same as the ones stating 200 W or even 1200 W and, with that, higher by a factor of 20 to 120.

Classic transformers are not produced anymore; new electronic devices have many features that the old ones did not have:

- Coupling for an easy installation
- Light in weight as they are 300 g instead of 4 kg
- Easy to obtain and still in production.

Even though the new type of transformer was designed and built with the purpose of replacing classic transformers in oil and gas heating systems, their application in the standards for the determination of safety characteristics was not explicitly allowed. With that, their usage was not common in the chemical safety field, or if it was, with mixed feelings.

There has been no comparative study so far about the two types of transformers, especially regarding the determination of different safety characteristics. This article may serve as an introduction to the new type of transformer, especially regarding its standards and apparatuses.

## 2. Materials and Methods

Two different transformers are compared in this work: a classic transformer (OP-TIMA German Lighting Component GmbH formerly May & Christe GmbH, Kleinostheim, Germany) and a new electronic transformer (Danfoss GmbH, Offenbach, Germany).

The safety characteristic maximum experimental safe gap (MESG) of hydrogen, methane, propane, ethylene, and acetylene were investigated with the new transformer and compared to the data in the literature. Additionally, the electric characteristics and calorimetric energy of both transformers were measured. The calorimeter was a bomb calorimeter with a signal of 37.7 $\mu$V/J, similar to the one described in [11]. The same electrodes for both transformers were placed inside the calorimeter. The length, cross section, and placement of the cables were identical for the electrical tests. For the calorimetric measurements, the cables between the transformer and the electrodes had a length of 3.2 m, a cross-section of 4 mm$^2$, an inductance of 4.7 $\mu$H, and a capacitance of 0.257 nF. For comparing the voltage of both transformers, the non-loaded secondary voltage of the classic and the electronic transformer was considered as the electric characteristic. This voltage was measured using a measuring setup consisting of a high-voltage probe (P6015A from Tektronix, Cologne, Germany) and an oscilloscope (WaveRunner 62Xi from Teledyne LeCroy Chestnut Ridge, New York, NY, USA). The high-voltage probe was a unit that consists of a resistive–capacitive voltage divider, a connecting cable, and a compensation box for adjustment. The probe had a divider ratio of 1:1000, a bandwidth of 75 MHz, and a maximum input voltage of 40 kV, peak. The probe was connected to the oscilloscope via the connection cable with the compensation box, and the voltage was recorded at a sampling rate of 5 GS/s. Figure 1 shows the schematic of the circuit for the voltage measurement of the classic transformer with center tap (dotted line) and the electronic transformer without center tap.

Each of the tested transformers were powered on the primary side by the input voltage $U_{grid,\sim}$ (see Table 1, primary voltage). On the secondary side, the probe was connected to the positive electrode of the classic and the electronic transformer as well as to the input channel of the oscilloscope (CH 1). In order to measure the non-loaded secondary voltage, the distance between the positive and negative electrode of the transformers was set high enough to prevent electric sparks from occurring between them.

**Table 1.** Technical data of the classic transformer and the new electronic transformer.

|  | Classic Transformer | Electronic Transformer |
|---|---|---|
| Manufacturer | May & Christe | Danfoss |
| Primary voltage | 230 V | 230 V |
| Secondary voltage (rms) | 15 kV | 15 kV |
| Secondary voltage (peak) | 21 kV | |
| Short-circuit current | 30 mA | 30 mA |
| Pulses per second | 100 | 50 |

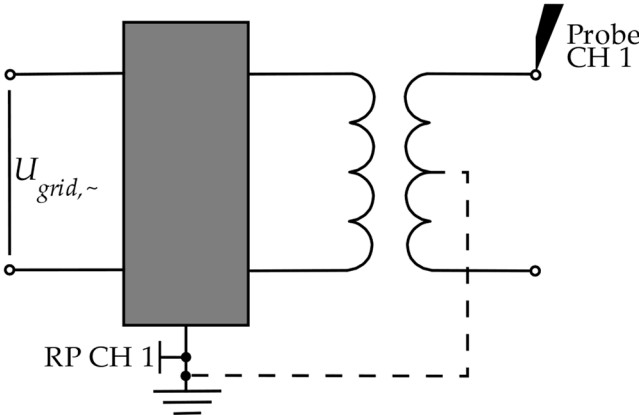

– –  Center tap of classic transformer
CH 1:  Channel 1 of oscilloscope
RP CH 1:  Reference point of channel 1

**Figure 1.** Schematic circuit diagram for the voltage measurement of the classic and the electronic transformer.

### 3. Results and Discussion

*3.1. Maximum Experimental Safe Gap*

The maximum experimental safe gap was investigated for hydrogen, methane, propane, ethylene, and acetylene with the new electronic transformer according to ISO 80079 [4]. The obtained values and the ones listed in a chemical database [12], as well as their differences, are displayed in Table 2.

**Table 2.** Values for the MESG for the different gases.

| Gas | Classic Transformer | Electronic Transformer | Difference |
| --- | --- | --- | --- |
| Hydrogen @ 26–29 mol% | **0.29 mm** | **0.29 mm** *0.30 mm* | 0.00 mm |
| Methane @ 8.6 mol% | **1.14 mm** | **1.14 mm** *1.15 mm* | 0.00 mm |
| Propane @ 4.1 mol% | **0.90 mm** | **0.91 mm** *0.92 mm* | 0.01 mm |
| Ethylene @ 6.7 and 6.9 mol% | **0.65 mm** | 0.63 mm *0.64 mm* | 0.02 mm |
| Acetylene @ 7.9 to 8.7 mol% | **0.37 mm** | **0.36 mm** *0.37 mm* | 0.01 mm |
| *Last breakthrough value* and **Safe Value** (=**MESG**) | | | |

The maximum experimental safe gap of the five different gases was the same value as the ones from the literature for two of the investigated gases, 10 μm higher for propane, 10 μm lower for acetylene, and 20 μm lower for ethylene. With that, it can be concluded that the transformers can be seen as equally suitable for the investigation of the MESG.

*3.2. Calorimetric Measurements*

To obtain a higher signal-to-noise ratio, the calorimetric measurements were obtained by triggering the ignition sources several times. Afterwards, the calorimetrically measured energy was divided by the overall spark duration (number of triggered ignitions and their duration) to display the power. The results are shown in Figure 2.

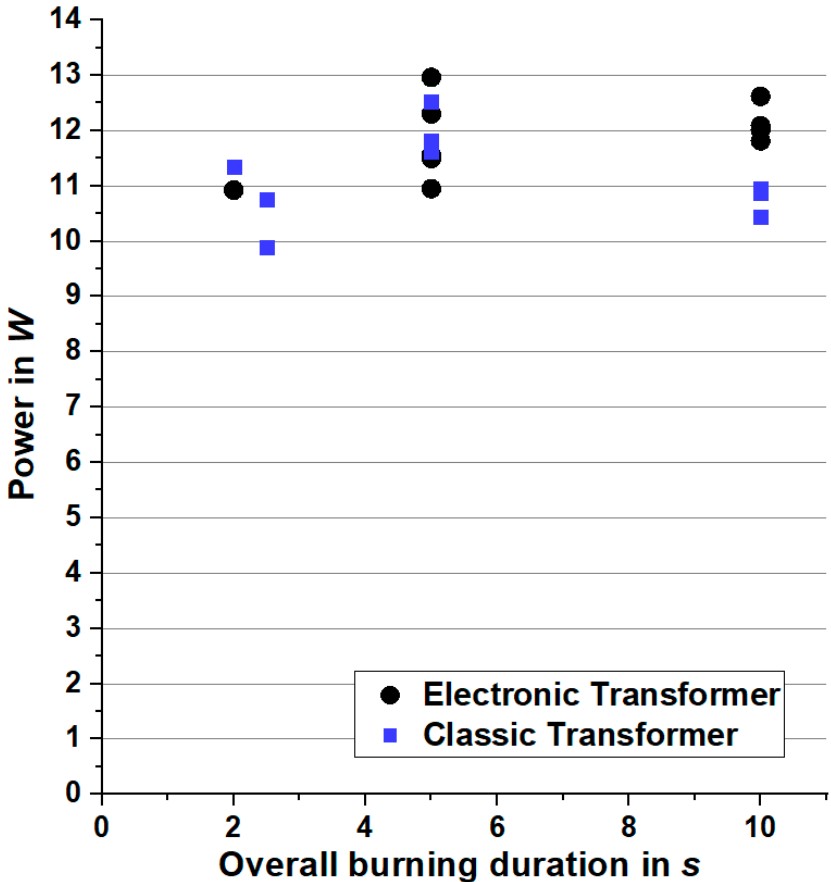

**Figure 2.** Calorimetrically measured power of the electronic transformer and two identical classic transformers.

The results of both types of transformers overlap at around 12 W on average. The scattering is also comparable with a maximum of ±2 W. With that, it can be said that the stated power of about 10 W in some standards [1,7,8,10] is right; those stating higher values are wrong and should be corrected.

### 3.3. Electrical Measurements

The voltage without load of both transformers was investigated. The voltage progress is displayed in Figures 3 and 4.

While the secondary voltage of the classic transformer has a sine shape with a peak value of around ±10 kV (from one pole to the ground/center tap, so double that value from pole to pole), the electronic transformer pulses only in the positive way and reaches a value of about 12 kV. In the detailed figure, it can also be seen that it is not a stable voltage over time for the electronic transformer as there are many high peaks with a frequency of about 14 kHz and intermediate small peaks with a frequency of 57 kHz. For these calculations, the peaks were simply counted within 20 ms and multiplied by 50.

The investigations into the maximum experimental safe gap of five gases as well as the calorimetrically measured net energy showed no significant difference (if at all) between the two types of transformers. The electrical investigation showed different behaviors of the voltage process and a different number of sparks per second, but these differences do not seem to affect the ignition behavior.

However, all the tests were performed under quiescent and ambient conditions. The behavior under extreme conditions, like elevated pressures and/or temperatures, unusual high vapor or dust loadings, high turbulence, or possible differences between the two types of transformers under such conditions, was not investigated and might lead to different results with the new type of transformer. Though, it is assumed that these conditions

might even lead to safer values of the safety characteristics with the electronic transformers because the voltage would stay at a higher peak value for a longer period. Further tests of the possible differences between the two transformers when investigating the explosion limits of gases, the current of both transformers when sparks occur, and the longest distance at which sparks still occur between the electrodes will be conducted in the future.

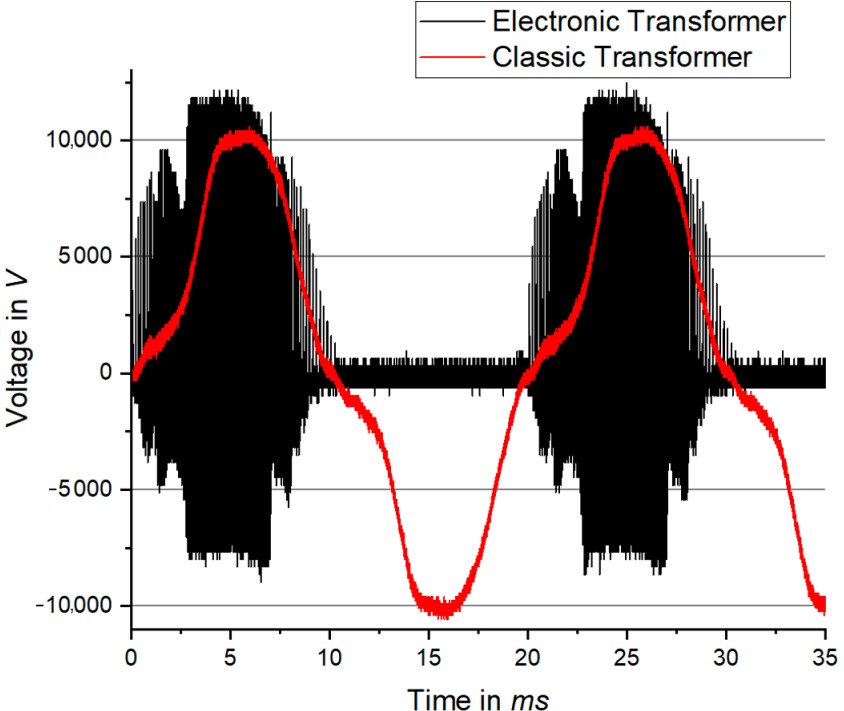

**Figure 3.** Voltage of the electronic and the classic transformer over time, two full pulses.

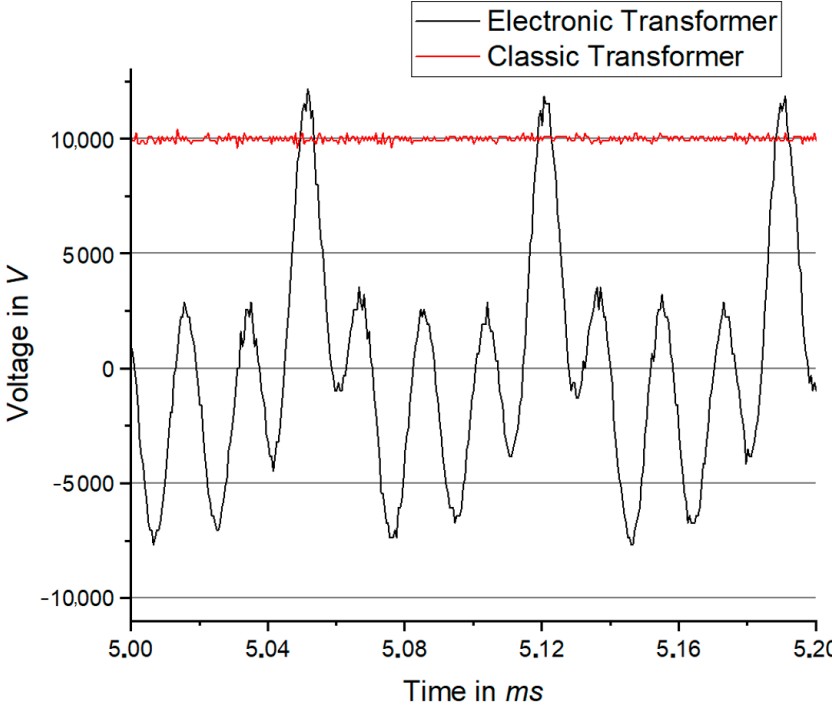

**Figure 4.** Voltage of the electronic and the classic transformer over time, in detail.

## 4. Conclusions

Two different transformers for the determination of safety characteristics were compared: a classic transformer and an electronic transformer. The comparison consisted of the determination of the MESG for five different gases, calorimetric measurements of the net energy that is introduced into the system, and investigations into the electrical behavior of both transformers.

Even though the electrical measurements revealed a different number of pulses with a different shape, the tests showed no differences in the ignition behavior between the two types of transformers. With these findings, it can be concluded that the new type of transformer can be applied in standard investigations and used in future work.

However, the behavior under extreme conditions (elevated pressures and/or temperatures, unusually high vapor or dust loadings, and high turbulence) was not investigated, which might lead to different results with the new type of transformer. Even though it is assumed that these conditions might lead to safer values of the safety characteristics with the electronic transformers, these values should be used with caution.

**Author Contributions:** Conceptualization, S.H.S. and S.Z.; methodology, S.H.S. and C.S.; validation, S.Z. and C.S.; formal analysis, G.R. and C.S.; investigation, G.R. and C.S.; resources, C.S. and S.Z.; data curation, G.R. and S.H.S.; writing—original draft preparation, S.H.S.; writing—review and editing, C.S.; visualization, S.H.S.; supervision, S.Z.; project administration, S.Z.; funding acquisition, S.Z. All authors have read and agreed to the published version of the manuscript.

**Funding:** This research received no external funding.

**Data Availability Statement:** The data presented in this study are available on request from the corresponding author.

**Acknowledgments:** The authors would like to thank Volkmar Schröder for help and advice on this complex topic.

**Conflicts of Interest:** The authors declare no conflict of interest.

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
