# Peer review of "A New Ignition Source for the Determination of Safety Characteristics of Gases"

_2673-5628, doi:10.3390/gases3030007_

Round 1

Reviewer 1 Report

In the manuscript "A new ignition source for the determination of safety characteristics of gases", the authors presents the investigation of five gases to assess two different transformers for the determination of safety characteristics. The manuscript is well-written, scientifically valid and logical in the presentation. Therefore, accepting the paper after a few minor revisions are made is suggested.

The following comments mainly deal with the presentation of the work—a few comments on the content.

·        Abstract. Please, improve the abstract by adding a few results.

·        The abbreviation list is not presented

·        Paragraph 1 "Introduction". It is suggested to double-check the sentences. Some of them result too long. Lines 19-26. “It is listed in the American and European standard 21for the determination of the explosion limits and the limiting oxygen concentration [1, 2, 3], the international standards for the determination of explosibility of gases [4] and dusts [5], the international standard for the determination of fire potential and oxidizing ability for the selection of cylinder valve outlets [6], the European standard for the determination of explosion points of flammable liquids [7] and of the maximum explosion pressure and the maximum rate of pressure rise of gases and vapors [8].

·        It is suggested to the authors further justify both the introduction section and the abstract regarding the novelty of this study with respect to the existing literature

·        Figure 2. Please, is suggested to reduce the y-scale to better show the results

Author Response

Dear Reviewer,

Thanks for your suggestions and invaluable remarks on the paper.

Here are your comments and my changes in green below.

Kind regards

Stefan Spitzer sen.

Reviewer 2 Report

In this paper, by comparing the electrical characteristics and the calorimetric energy of the classical transformer and the new electronic transformer, as well as the safety characteristic maximum experimental safe gap (MESG) of hydrogen, methane, propane, ethylene and acetylene under the two transformer conditions, it can be concluded that the new type of transformer can be applied in standard investigations and used in future work.

Innovation: The new electronic transformer is selected as the research object, which is practical and provides a new ignition source reference for the determination of gas safety characteristics.

However, there are still some problems that need to be modified in this paper, as follows:

1. In 3.2 Calorimetric measurements, only the chart is given, and the content displayed in the chart is not explained. Please explain the chart.

2. In 3.3 Electrical measurements, In the detail figure it can also be seen that it is not a stable voltage over time for the electronic transformer but many high peaks with a frequency of about 14 kHz and intermediate small peaks with a frequency of 57 kHz. Please add the calculation method of 14 kHz and 57 kHz.

3. Pleas add some related references.

4. In Table 2, to be consistent with other data formats, please roughen the 0.63 corresponding to ethylene.

Author Response

Dear Reviewer,

Thanks for your suggestions and invaluable remarks on the paper.

Here are your comments and my changes/answers in green below.

Kind regards

Stefan Spitzer sen.
